# The Impact of Nutritional Markers and Dietary Habits on the Bioimpedance Phase Angle in Older Individuals

**DOI:** 10.3390/nu15163599

**Published:** 2023-08-17

**Authors:** Shintaro Kajiyama, Naoko Nakanishi, Shinta Yamamoto, Takahiro Ichikawa, Takuro Okamura, Yoshitaka Hashimoto, Noriyuki Kitagawa, Masahide Hamaguchi, Michiaki Fukui

**Affiliations:** 1Department of Endocrinology and Metabolism, Graduate School of Medical Science, Kyoto Prefectural University of Medicine, Kyoto 602-8566, Japan; 2Department of Diabetes and Endocrinology, Matsushita Memorial Hospital, Moriguchi 570-8540, Japan; 3Kameoka Municipal Hospital, Kyoto 621-8585, Japan

**Keywords:** phase angle, serum cholinesterase, total energy intake, protein intake

## Abstract

Low phase angle (PhA), as determined via bioelectrical impedance analysis, reflects unhealthy aging and mortality. In this study, we assessed whether nutritional status, including serum nutritional markers and dietary habits, is related to PhA in older individuals. We recruited 212 participants (aged ≥ 65 years) who underwent medical health checkups. PhA was measured using a multi-frequency impedance body composition analyzer. Habitual food and nutrient intake was evaluated using a brief, self-administered diet history questionnaire. Low PhA values were defined as ≤4.95 in males and ≤4.35 in females. Males with low PhA had poor exercise habits (*p* = 0.0429) and a lower body mass index (*p* = 0.0024). PhA was significantly correlated with serum cholinesterase levels, a nutritional status marker (r = 0.3313, *p* = 0.0004 in males; r = 0.3221, *p* = 0.0070 in females). The low-PhA group had significantly lower total energy and carbohydrate intake per ideal body weight (IBW) than the high-PhA group in males (total energy intake:30.2 ± 9.8 and 34.5 ± 9.3 kcal/kg/day, *p* = 0.0307; carbohydrate intake:15.2 ± 4.9 and 18.0 ± 5.8 kcal/kg/day, *p* = 0.0157). Total energy intake per IBW (adjusted odds ratio [95% confidence interval], 0.94 [0.89–1.00] per 1 kcal/kg/day increase) was independently associated with a low PhA in males. Our study revealed that lower total energy intake independently impacted low PhA in older males.

## 1. Introduction

Bioelectrical impedance (BI) analysis (BIA) is used to estimate body components by applying a weak electric current to the body and measuring impedance. This non-invasive and inexpensive method is employed to assess a patient’s body composition in various fields, including routine medical care and sports medicine [1]. BI utilizes whole-body measurements to classify and monitor hydration and cell mass, focusing on raw bioelectric parameters, such as resistance (R) and reactance (Xc). Here, R represents the opposition to the flow of low-level alternating current owing to ionic fluids, and Xc represents the delay in current entry into cells related to cell membranes and cell interfaces. The phase angle (PhA) was calculated as follows: PhA (°) = [−arc tangent (Xc/R) × 180°/π] [2]. PhA represents the resistance of cell membranes, somatic cell volume [3,4], and intra- and extracellular fluid distribution [5]. A high PhA value typically indicates better cellular health and integrity, whereas a low PhA value reflects structural damage to cell membranes or reduced cell density, indicating compromised cellular function [6].

Typically, PhA values are lower in females than in males and decrease with increasing age [7,8]. In addition, Asians were found to exhibit lower PhA values compared to Western populations [5], with healthy Asians presenting PhA values of 5.1 ± 0.6 in men and 4.6 ± 0.5 in women aged ≥65 years [9].

The PhA has been deemed a substantial prognostic factor in the care of critically ill patients [10]. The correlation between a low PhA and mortality has been well documented, particularly in patients with kidney disease and cancer [11]. Moreover, older individuals with a low PhA reportedly exhibit an increased risk of key features of unhealthy aging, such as sarcopenia, frailty, and mortality [12,13]. A Japanese study assessed individuals aged ≥65 years and found that low PhA categorized by a cutoff value (male, ≤4.95°; female, ≤4.35°) was independently associated with an increased risk of incident disability (hazard ratio (HR) = 1.95, 95% confidence interval (CI) = 1.37–2.78) [14]. Accordingly, a low PhA may reflect unhealthy aging and mortality in patients with degenerative diseases and healthy subjects.

Recent studies have indicated that PhA can be influenced by physical activity level [15], and a meta-analysis revealed that resistance training promotes an increase in PhA [16]. However, nutritional status, including the dietary habits associated with PhA in healthy subjects, remains poorly understood. Herein, we assessed older individuals who underwent a health checkup with documented BIA values, blood assessment results, and dietary history; we also examined whether serum nutritional markers and dietary habits are related to a low PhA in older individuals who underwent a medical health checkup.

## 2. Materials and Methods

### 2.1. Study Design

We conducted a cohort analysis of participants who underwent medical health checkups at Kameoka Municipal Hospital. The checkup results were saved in a database, and personally identifiable information was removed. This longitudinal cohort analysis was conducted using the HOZUGAWA database. The study was approved by the Ethics Committee of Kyoto Prefectural University of Medicine (approval no. ERB-C-1503) and performed in accordance with the Declaration of Helsinki. We recruited 212 participants aged ≥65 years who underwent medical health checkups between January 2018 and March 2021. Informed consent was obtained from all participants involved in the study.

### 2.2. Data Collection

We evaluated age, weight, body mass index (BMI), blood pressure, lifestyle factors, such as smoking history, regular exercise, skipping breakfast, eating speed, and sleep quality, and blood test results, including aspartate aminotransferase (AST), alanine aminotransferase (ALT), γ-glutamyl transpeptidase (γ-GTP), creatinine, fasting blood glucose, hemoglobin A1c (HbA1c), triglyceride, high-density lipoprotein (HDL) cholesterol, and total protein levels. Furthermore, we determined the serum total cholesterol, albumin, cholinesterase, and total lymphocyte counts, which are well-known sensitive markers of nutritional status [17,18]. Blood tests were conducted in the morning after overnight fasting. The BMI was defined as body weight (kg)/height squared (m^2^). The estimated glomerular filtration rate (eGFR) was calculated using the Japanese Society of Nephrology equation, as follows: eGFR = 194 × serum creatinine^−1.094^ × age^−0.287^ × 0.739 (for females) (mL/min/1.73 m^2^). Participants were questioned regarding their exercise habits, and those who exercised two or more times a week were categorized as those who exercised regularly.

The body composition of the participants was evaluated using segmental multifrequency bioelectrical impedance analysis (InBody 720, InBody Japan, Tokyo, Japan). The system separately measured the impedance of the participant’s right arm, left arm, trunk, right leg, and left leg at six different frequencies (1, 5, 50, 250, 500, and 1000 kHz) for each body segment. The analyzer can directly measure fat-free mass (FFM), soft lean mass (SLM), and body fat mass (BFM) without using an equation for correction. Whole-body, appendicular fat mass and lean soft tissue mass measured using InBody 720 were found to correlate well with those measured by dual-energy X-ray absorptiometry and have been validated previously [19]. The examination was conducted in the morning after an overnight fast. The skeletal muscle mass index (SMI; kg/m^2^) was defined as appendicular lean mass (kg)/height squared (m^2^). The ideal body weight (IBW) was defined as 22 × patient height squared (m^2^) [20]. The percentages of body fat mass and skeletal muscle mass were calculated as body fat mass and skeletal muscle mass, respectively, divided by the body weight. Whole-body raw values of R and Xc at 50 kHz were measured using InBody 720, and PhA values were obtained by calculating PhA (°) = [−arc tangent (Xc/R) × 180°/π] [2,3].

A brief self-administered diet history questionnaire (BDHQ) was used to evaluate habitual food and nutrient intake. The BDHQ estimates dietary intake and variations in 58 food items over the past month using an algorithm based on the Standard Tables of Food Composition 2010 in Japan (Standard Tables of Food Composition in Japan; Ministry of Education, Culture, Sports, Science and Technology, Tokyo, Japan, 2010). The BDHQ has been validated previously [21,22]. Carbohydrate (g/day), total protein (g/day), animal protein (g/day), vegetable protein (g/day), fat (g/day), dietary fiber (g/day), and alcohol consumption (g/day) were calculated. The energy intake of carbohydrates, proteins, and fats was calculated as carbohydrate (g/day) × 4, protein (g/day) × 4, and fat (g/day) × 9. Energy intake per IBW was defined as energy intake (kcal/day) divided by IBW (kg). Alcohol consumption was evaluated according to the amount and type of alcohol consumed per week during the previous month, and the mean ethanol intake per week was estimated.

Given that PhA values significantly differ between males and females (5.28 ± 0.59° vs. 4.69 ± 0.46°, *p* < 0.0001), all analyses were performed separately for males and females. We defined ≤4.95° and ≤4.35° as low PhA in males and females, respectively; these values were based on a previous report where these cutoff values were associated with an increased risk of incident disability [14]. Characteristics of the high- and low-PhA groups were compared. In addition, we investigated the correlation between the PhA value and each factor. Furthermore, we evaluated the independent factors associated with low PhA. Covariates included age, exercise habits, and total energy intake.

### 2.3. Statistical Analysis

Continuous values are presented as mean ± standard deviation (SD or median (interquartile range)), whereas categorical values are presented as n (%). Student’s *t*-test or Wilcoxon signed-rank test was used to analyze statistical differences in continuous variables. The Pearson correlation coefficient was used to examine the correlation between the PhA values and the examined factors. Logistic regression analysis was performed to calculate the unadjusted and adjusted odds ratios (ORs) and 95% CIs for low phA. All statistical analyses were performed using the JMP Pro 15 software (https://www.jmp.com/en_us/software/predictive-analytics-software.html, accessed on 1 August 2023). A *p* value less than 0.05 was considered statistically significant.

## 3. Results

A total of 212 participants aged ≥65 years (130 males and 82 females) who underwent a medical health check-up at Kameoka Municipal Hospital were enrolled in the current study. The characteristics of the participants and comparisons between the high- and low-PhA groups for males and females are shown in Table 1 and Table 2, respectively. The correlations between PhA and participant characteristics are presented in Table 3.

Older male and female participants exhibited lower PhA values. The results of the correlation analysis revealed that the PhA was negatively correlated with age in both males and females (r = −0.5286, *p* < 0.0001 in males; r = −0.4051, *p* = 0.0006 in females). Males with a low PhA had a lower body mass index than those with a high PhA (*p* = 0.0024). PhA positively correlated with BMI in males (r = 0.3207, *p* = 0.0006), with no correlation between PhA and BMI in females.

Regarding lifestyle assessment, males with low PhA had poor exercise habits (*p* = 0.0429).

According to the body composition analysis, the low-PhA group had significantly lower skeletal muscle mass and SMI than the high-PhA group. Correlation analysis revealed that PhA significantly correlated with skeletal muscle mass and SMI (r = 0.4241, *p* < 0.0001 in males; r = 0.3562, *p* = 0.0027 in females). Among female participants, the low-PhA group had a lower percentage of skeletal muscle mass than the high-PhA group; however, no difference was detected among male participants.

Considering the serum markers of nutritional status, the low-PhA group exhibited lower serum albumin and cholinesterase levels than the high-PhA group. Correlation analysis revealed that PhA was positively correlated with serum cholinesterase levels in both males and females (r = 0.3313, *p* = 0.0004 in males; r = 0.3221, *p* = 0.0070 in females).

A comparison of the habitual diet between the high-PhA and low-PhA groups is shown in Figure 1b (males) and Figure 1c (females). In males, the total energy intake per IBW was significantly lower in the low-PhA group than in the high-PhA group (30.2 ± 9.8 and 34.5 ± 9.3 kcal/kg/day, *p* = 0.0307). Likewise, carbohydrate intake per IBW was significantly lower in the low-PhA group than that in the high-PhA group (15.2 ± 4.9 and 18.0 ± 5.8 kcal/kg/day, *p* = 0.0157). PhA correlated with total energy intake per IBW (r = 0.1828, *p* = 0.0537) and carbohydrate intake per IBW (r = 0.2000, *p* = 0.0362); however, there was no correlation between PhA and protein intake. In females, the correlation coefficient of PhA with total energy was 0.288 (*p* = 0.0164), and the total energy per IBW was 0.222 (*p* = 0.0667).

Table 4 presents the ORs for low PhA values. In males, age (OR (95% CI), 1.22 (1.09–1.37) per 1-year increase) and total energy intake per IBW (OR (95% CI), 0.94 (0.89–1.00) per 1 kcal/kg/day increase) were independently associated with low PhA values. In females, age (OR (95% CI), 1.22 (1.07–1.39) per 1-year increase) was independently associated with low PhA values.

## 4. Discussion

In the present study, we explored the factors associated with a low PhA in older individuals who had undergone a medical checkup. Age is an important factor associated with PhA [7,8]. Both older males and females exhibited low phA levels. This study also revealed that males with low PhA values exhibited poor exercise habits. These findings are consistent with those of a previous study [15], which demonstrated a positive association between physical activity and PhA.

This finding is consistent with those of previous studies showing that PhA is associated with SMI [23] or sarcopenia [24,25]. However, the association between the PhA and BMI remains controversial. A previous study assessing participants with obesity found that individuals with BMI ≥ 28 had higher PhA values than those with BMI < 28 [26]. In contrast, another study found no correlation between PhA and BMI [27]. In the current study, PhA was positively correlated with BMI in males, whereas no correlation was observed between PhA and BMI in females. We also observed that females, but not males, in the low-PhA group had a low percentage of skeletal muscle mass.

Regarding nutritional indices, the low-PhA group had significantly lower serum cholinesterase levels than the high-PhA group, and cholinesterase levels correlated well with PhA. In addition, serum albumin levels were correlated with PhA values. An association between serum albumin levels and PhA has been previously reported in hospitalized patients [28] and patients with advanced cancer [29]; however, an association between PhA and nutritional indices in healthy individuals is yet to be established. Serum cholinesterase reflects the protein synthesis capacity of the liver and is frequently employed as an indicator of liver function; it is also considered a useful nutritional indicator [30]. Our study showed a significant correlation between PhA values and serum cholinesterase levels in older individuals who underwent medical health checkups, indicating that PhA reflects the nutritional status of healthy older individuals.

However, the association between dietary habits and PhA has been poorly explored. The intake of red meat and chicken has been correlated with PhA [23] in patients with type 2 diabetes. In contrast, Unterberger et al. detected no relationship between protein intake and PhA in older individuals using 24 h dietary recalls through personal interviews [31]. In addition, we found no correlation between protein intake and PhA levels in males or females. To help older individuals (>65 years) maintain and regain lean body mass and function, the PROT-AGE study group recommends an average protein intake of at least 1.0–1.2 g per kilogram of body weight per day [32,33]. In a previous study investigating the nutritional status of predialysis patients, participants in the lowest tertile of protein intake had a low PhA [34]. In the present study, the average protein intake in the low-PhA group was 1.2 g/kg/day in males and 1.4 g/kg/day in females. Generally, extremely low protein intake may result in low PhA; however, in the current study, protein intake did not affect PhA in individuals who consumed a certain amount of protein.

Our study indicated that low carbohydrate intake contributes to low total energy intake in males. Logistic regression analysis revealed that the total energy intake was an independent factor affecting low PhA in males. Previous studies have shown that insufficient energy intake is associated with muscle mass loss in older adults with type 2 diabetes [35] and that low energy intake leads to a high mortality risk in older people with diabetes [36]. The ESPEN Guidelines on Nutrition in Geriatrics also state that restrictive diets are not recommended for overweight older adults [37]. A meta-analysis suggested that nutritional interventions in patients with cancer could increase their PhA values [38]. We suggest that preventing a shortage of carbohydrates and ensuring a sufficient intake of total calories may contribute to an increased PhA and promote healthy aging among healthy older males.

There was no independent relationship between total energy intake and PhA in females in this study, which could be attributed to females with sufficient protein intake in the current study being less susceptible to the effects of total energy intake.

Our study had some limitations. First, owing to the cross-sectional design, the results did not allow for causal associations. Second, a relatively small number of participants, especially women, were included. We cannot rule out the possibility that the significant differences detected in the principal findings in men but not in women could be attributed to insufficient sample size. Despite these limitations, our study provides valuable insights into the association between PhA and nutritional status in older individuals. The comprehensive list of analyzed markers and robust findings presented herein provide a foundation for further research in this field.

Furthermore, it is crucial to acknowledge that PhA values exhibit variability attributed to race, gender, age, and body size. As a result, meticulous attention is essential when interpreting these outcomes. As an alternative approach, Bioelectrical Impedance Vector Analysis (BIVA) employs the normalization of impedance parameters R and Xc with respect to height, utilizing a bivariate vector on the RXc graph [39]. BIVA provides a qualitative assessment of soft tissue that remains unaffected by body size, offering valuable insights into hydration status and the integrity of cellular mass [6]. Another innovative strategy involves standardizing PhA using reference values adjusted for age, sex, and BMI. Standardized values potentially provide a more reliable indication of nutritional or health status in comparison to absolute PhA values [6,39].

## 5. Conclusions

Our study examined the factors related to PhA in older individuals who underwent a medical health checkup and found that serum cholinesterase levels were significantly correlated with PhA as a nutritional index. By assessing lifestyle and dietary habits, our study showed that lower total energy intake was an independent factor affecting low PhA in older males.

## Figures and Tables

**Figure 1 nutrients-15-03599-f001:**
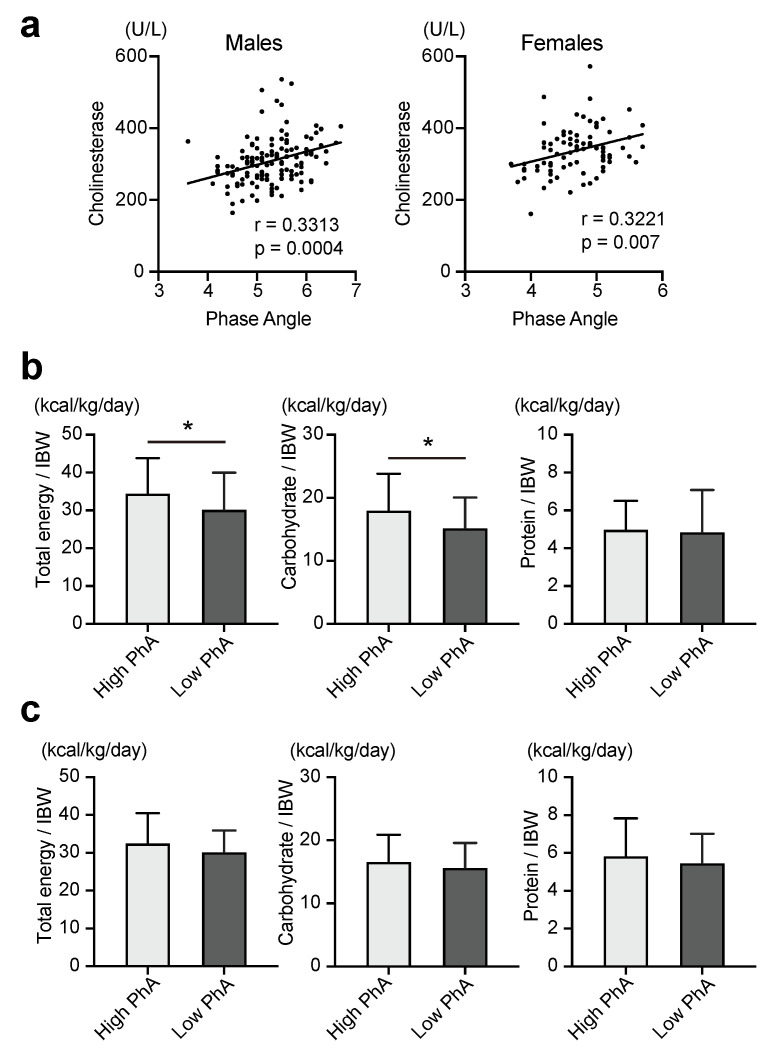
(**a**) Correlation between PhA values and levels of serum cholinesterase in males and females where r denotes the correlation coefficient. The *p*-values were calculated using the Pearson correlation coefficient analysis; (**b**) comparison of total calories, carbohydrate intake per IBW and protein intakes per IBW between the high- and low-PhA groups in males. The *p*-value was calculated using the Student’s *t*-test. Data are presented as mean ± standard deviation (SD). * *p* < 0.05; (**c**) comparison of total calories, carbohydrate intake per IBW, and protein intakes per IBW between the high- and low-PhA groups in females. The *p*-value was calculated using the Student’s *t*-test. Data are presented as mean ± standard deviation (SD). * *p* < 0.05.

**Table 1 nutrients-15-03599-t001:** Comparison of clinical characteristics between high- and low-phase-angle (PhA) groups in males.

Characteristics	All n = 130	High PhA n = 91	Low PhA n = 39	*p*
Age (years)	72.6 (5.0)	71.2 (4.4)	75.3 (5.2)	<0.0001
BMI (kg/m^2^)	23.1 (3.1)	23.5 (2.9)	21.8 (3.1)	0.0024
Habit of smoking, current smoker, n (%)	14 (12.7)	12 (14.2)	3 (8.3)	0.3774
Habit of exercise, yes, n (%)	68 (56.2)	53 (61.6)	15 (41.7)	0.0429
Skipping breakfast, yes, n (%)	6 (5.0)	3 (3.5)	3 (8.6)	0.2429
Fast eating speed, yes, n (%)	64 (52.9)	47 (55.3)	17 (47.2)	0.4161
Getting a good sleep, yes, n (%)	100 (82.6)	71 (82.6)	29 (82.9)	0.9686
Systolic Blood Pressure (mmHg)	136.7 (17.1)	137.5 (16.8)	134.1 (18.4)	0.3194
Diastolic Blood Pressure (mmHg)	83.0 (11.0)	84.3 (10.0)	79.6 (12.7)	0.024
AST (mg/dL)	23.6 (7.1)	23.7 (6.9)	23.4 (7.7)	0.8075
ALT (mg/dL)	22.0 (10.6)	22.4 (9.9)	20.6 (12.4)	0.3542
γ-GTP (mg/dL)	43.7 (43.2)	46.9 (48.2)	36.3 (29.2)	0.2027
Creatinine	1.2 (2.1)	1.2 (2.5)	1.0 (0.2)	0.5066
eGFR (mL/min/1.73 m^2^)	59.8 (12.8)	59.4 (12.8)	60.2 (13.0)	0.7223
Fasting Blood Glucose (mg/dL)	111.0 (19.3)	111.1 (19.9)	111.0 (18.5)	0.9563
Hemoglobin A1c (%)	6.0 (0.7)	5.9 (0.7)	6.1 (0.7)	0.2089
Triglyceride	105.8 (47.0)	109.3 (48.4)	96.4 (43.7)	0.1533
HDL cholesterol	63.7 (18.9)	63.5 (19.3)	64.5 (19.0)	0.7904
Total cholesterol	197.2 (30.6)	200.3 (29.9)	190.1 (32.1)	0.0819
Total protein (g/dL)	7.0 (0.4)	7.0 (0.4)	6.9 (0.4)	0.3683
Albumin (g/dL)	4.3 (0.4)	4.3 (0.4)	4.2 (0.3)	0.1524
Cholinesterase (U/L)	308.0 (65.2)	320.3 (68.2)	279.5 (48.2)	0.0009
Total lymphocyte count (10^2^/uL)	29.8 (7.8)	30.2 (8.2)	28.6 (6.8)	0.2796
**The body composition**				
Body fat mass (kg)	15.4 (5.3)	15.5 (4.9)	15.0 (6.1)	0.4961
Percentage of body fat mass	23.9 (6.2)	23.7 (5.3)	23.9 (8.0)	0.8114
Skeletal muscle mass (kg)	20.0 (2.7)	20.5 (2.5)	19.0 (2.9)	0.0048
Skeletal muscle mass index (kg/m^2^)	7.3 (0.7)	7.5 (0.7)	6.9 (0.7)	<0.0001
Percentage of skeletal muscle mass (%)	31.9 (3.0)	31.9 (2.5)	31.7 (3.8)	0.7722
**Habitual diet intake**				
Total energy intake (kcal/day)	1989.2 (564.7)	2057.1 (568.9)	1826.0 (545.0)	0.0487
Total energy intake per IBW (kcal/kg/day)	33.0 (9.7)	34.5 (9.3)	30.2 (9.8)	0.0307
Carbohydrate intake (kcal/day)	1031.5 (334.2)	1081.1 (348.3)	918.4 (286.8)	0.0189
Carbohydrate intake per IBW (kcal/kg/day)	17.1 (5.6)	18.0 (5.8)	15.2 (4.9)	0.0157
Protein intake (kcal/day)	398.8 (100.9)	299.3 (89.3)	292.6 (126.6)	0.7489
Animal protein intake (g/day)	43.6 (21.0)	42.6 (17.5)	44.5 (27.9)	0.6575
Vegetable protein intake (g/day)	31.1 (8.8)	32.3 (8.9)	28.6 (8.5)	0.0478
Protein intake per IBW (kcal/kg/day)	5.0 (1.7)	5.0 (1.5)	4.8 (2.2)	0.7035
Protein intake per IBW (g/kg/day)	1.2 (0.4)	1.2 (0.4)	1.2 (0.5)	0.7035
Fat intake (kcal/day)	441.7 (137.1)	448.8 (137.9)	419.0 (136.2)	0.2967
Fat intake per IBW (kcal/kg/day)	7.3 (2.3)	7.5 (2.2)	6.9 (2.5)	0.2763
Dietary fiber intake (g/day)	12.5 (4.4)	12.7 (4.5)	12.0 (4.4)	0.4832
Alcohol consumption (g/day)	14.4 (1.9–28.7)	15.9 (1.3–28.8)	10.3 (2.5–29.4)	0.7189

Data are presented as n (%), mean ± standard deviation (SD), or median (25th and 75th percentiles). AST, aspartate aminotransferase; ALT, alanine aminotransferase; eGFR, estimated glomerular filtration rate; HDL, high-density lipoprotein; IBW, ideal body weight. The *p*-values were calculated using the Student’s *t*-test or Wilcoxon signed-rank test.

**Table 2 nutrients-15-03599-t002:** Comparison of clinical characteristics between high- and low-phase-angle (PhA) groups in females.

Characteristics	All n = 82	High PhA n = 59	Low PhA n = 23	*p*
Age (years)	71.3 (4.9)	69.9 (3.6)	74.5 (6.4)	0.0001
BMI (kg/m^2^)	22.4 (3.9)	22.8 (4.0)	22.0 (3.4)	0.4045
Habit of smoking, current smoker, n (%)	3 (4)	3 (4)	0 (0)	0.2243
Habit of exercise, yes, n (%)	44 (55)	34 (58.6)	10 (45.5)	0.2905
Skip breakfast, yes, n (%)	2 (2.5)	2 (3.5)	0 (0)	0.3777
Fast eating speed, yes, n (%)	44 (55.0)	30 (51.7)	14 (63.6)	0.3389
Getting a good sleep, yes, n (%)	58 (73.4)	43 (74.1)	15 (71.4)	0.8097
Systolic Blood Pressure (mmHg)	134.9 (16.3)	134.5 (16.7)	136.9 (15.7)	0.5532
Diastolic Blood Pressure (mmHg)	77.9 (10.9)	79.2 (11.5)	75.8 (8.7)	0.2054
AST (mg/dL)	23.2 (5.2)	23.1 (4.7)	23.1 (6.0)	0.9878
ALT (mg/dL)	19.4 (8.3)	20.0 (7.7)	17.1 (9.0)	0.1492
γ-GTP (mg/dL)	24.4 (18.3)	26.0 (20.4)	20.5 (11.7)	0.2317
Creatinine	0.7 (0.1)	0.7 (0.1)	0.6 (0.1)	0.0115
eGFR (mL/min/1.73 m^2^)	64.3 (10.8)	62.9 (10.6)	68.3 (10.9)	0.043
Fasting Blood Glucose (mg/dL)	103.8 (15.3)	104.2 (16.3)	103.1 (13.5)	0.7728
Hemoglobin A1c (%)	5.9 (0.5)	5.9 (0.5)	5.9 (0.3)	0.6006
Triglyceride	89.8 (36.2)	88.5 (36.4)	93.7 (37.6)	0.5685
HDL cholesterol	76.4 (15.3)	77.3 (14.7)	72.1 (15.3)	0.1576
Total cholesterol	218.0 (30.0)	217.0 (27.8)	215.6 (32.1)	0.8461
Total protein (g/dL)	7.1 (0.4)	7.1 (0.3)	7.0 (0.4)	0.1012
Albumin (g/dL)	4.3 (0.3)	4.3 (0.3)	4.2 (0.3)	0.0748
Cholinesterase (U/L)	335.6 (65.7)	350.5 (60.7)	304.9 (65.1)	0.0036
Total lymphocyte count (10^2^/uL)	32.5 (8.5)	32.9 (7.3)	30.3 (10.6)	0.2105
**The body composition**				
Body fat mass (kg)	17.2 (7.1)	17.5 (0.9)	17.0 (1.5)	0.7627
Percentage of body fat mass	31.5 (8.1)	31.3 (8.4)	32.9 (7.0)	0.4097
Skeletal muscle mass (kg)	14.0 (2.5)	14.4 (2.5)	12.9 (2.3)	0.0144
Skeletal muscle mass index (kg/m^2^)	5.9 (0.8)	6.1 (0.7)	5.6 (0.7)	0.0035
Percentage of skeletal muscle mass (%)	26.7 (3.3)	27.1 (3.4)	25.5 (2.8)	0.0428
**Habitual diet intake**				
Total energy intake (kcal/day)	1651.3 (369.7)	1679.3 (388.7)	1540.1 (286.3)	0.1455
Total energy intake per IBW (kcal/kg/day)	32.1 (7.5)	32.5 (7.9)	30.2 (5.7)	0.2328
Carbohydrate intake (kcal/day)	851.5 (221.8)	860.2 (222.4)	798.9 (201.1)	0.2823
Carbohydrate intake per IBW (kcal/kg/day)	16.5 (4.3)	16.6 (4.3)	15.7 (3.9)	0.3869
Protein intake (kcal/day)	269.2 (89.6)	299.3 (94.2)	278.0 (69.5)	0.3552
Animal protein intake (g/day)	46.5 (19.2)	47.5 (20.0)	42.3 (16.1)	0.2907
Vegetable protein intake (g/day)	27.6 (6.9)	27.3 (7.0)	27.2 (6.4)	0.9715
Protein intake per IBW (kcal/kg/day)	5.8 (1.9)	5.8 (2.0)	5.5 (1.5)	0.4685
Protein intake per IBW (g/kg/day)	1.4 (0.5)	1.5 (0.5)	1.4 (0.4)	0.4685
Fat intake (kcal/day)	418.5 (125.5)	428.5 (133.3)	391.7 (27.3)	0.2641
Fat intake per IBW (kcal/kg/day)	8.1 (2.6)	8.3 (2.7)	7.6 (2.1)	0.3379
Dietary fiber intake (g/day)	12.1 (3.9)	11.8 (3.9)	12.5 (3.6)	0.5153
Alcohol consumption (g/day)	0.0 (0.0–0.8)	0.0 (0.0–8.2)	0.0 (0.0–0.3)	0.938

Data are presented as n (%), mean ± standard deviation (SD), or median (25th and 75th percentiles). AST, aspartate aminotransferase; ALT, alanine aminotransferase; eGFR, estimated glomerular filtration rate; HDL, high-density lipoprotein; IBW, ideal body weight. The *p*-values were calculated using the Student’s *t*-test or Wilcoxon signed-rank test.

**Table 3 nutrients-15-03599-t003:** Correlation between phase angle (PhA) and other factors.

Characteristics	Men		Women	
	r	*p*	r	*p*
Age (years)	−0.5286	<0.0001	−0.4051	0.0006
BMI (kg/m^2^)	0.3207	0.0006	0.0774	0.5274
Systolic Blood Pressure (mmHg)	0.0608	0.5302	−0.0694	0.5739
Diastolic Blood Pressure (mmHg)	0.1782	0.0638	0.0204	0.8686
AST (mg/dL)	0.1157	0.231	0.0265	0.8301
ALT (mg/dL)	0.2381	0.0126	0.1767	0.1496
γ-GTP (mg/dL)	0.1598	0.0971	0.0968	0.4324
Creatinine	−0.0137	0.8872	0.191	0.1187
eGFR (mL/min/1.73 m^2^)	0.0296	0.76	−0.1504	0.2209
Fasting Blood Glucose (mg/dL)	0.0377	0.697	0.1648	0.1793
Hemoglobin A1c (%)	−0.0245	0.8004	0.214	0.0797
Triglyceride	0.1557	0.106	−0.1034	0.2841
HDL cholesterol	−0.0644	0.506	0.1318	0.4013
Total cholesterol	0.1076	0.2653	−0.0316	0.2841
Total protein (g/dL)	0.1379	0.151	0.1949	0.1085
Albumin (g/dL)	0.102	0.2889	0.2988	0.0126
Cholinesterase (U/L)	0.3313	0.0004	0.3221	0.007
Total lymphocyte count (/uL)	0.1152	0.2308	0.1343	0.2713
**The body composition**				
Body fat mass (kg)	0.0759	0.4328	−0.0316	0.7982
Percentage of body fat mass (%)	−0.0231	0.8112	−0.137	0.2653
Skeletal muscle mass (kg)	0.3946	<0.0001	0.3534	0.0031
Skeletal muscle mass index (kg/m^2^)	0.4241	<0.0001	0.3562	0.0027
Percentage of skeletal muscle mass (%)	−0.002	0.8364	0.2524	0.0379
**Habitual diet intake**				
Total energy intake (kcal/day)	0.1303	0.1747	0.288	0.0164
Total energy intake per IBW (kcal/kg/day)	0.1828	0.0537	0.222	0.0667
Carbohydrate intake (kcal/day)	0.1747	0.068	0.247	0.0408
Carbohydrate intake per IBW (kcal/kg/day)	0.2	0.0362	0.1967	0.1052
Protein intake (kcal/day)	−0.0293	0.7613	0.161	0.0608
Animal protein intake (g/day)	−0.1032	0.2855	0.1581	0.1978
Vegetable protein intake (g/day)	0.161	0.0945	0.0782	0.5264
Protein intake per IBW (kcal/kg/day)	−0.0093	0.9231	0.1085	0.3748
Fat intake (kcal/day)	0.0248	0.7972	0.2269	0.0608
Fat intake per IBW (kcal/kg/day)	0.0422	0.6615	0.1804	0.1381
Dietary fiber intake (g/day)	0.0255	0.792	−0.0032	0.9791

The correlation coefficient is denoted by r. AST, aspartate aminotransferase; ALT, alanine aminotransferase; eGFR, estimated glomerular filtration rate; HDL, high-density lipoprotein; IBW, ideal body weight. The *p*-values were calculated using Pearson’s correlation coefficients.

**Table 4 nutrients-15-03599-t004:** Odds ratio of low phase angle (low PhA).

	Men				Women			
	Unadjusted		Adjusted		Unadjusted	Adjusted		
	OR (95% CI)	*p*	OR (95% CI)	*p*	OR (95% CI)	*p*	OR (95% CI)	*p*
Age	1.2		1.22		1.22		1.22	
(per 1 year increase)	(1.10–1.31)	<0.0001	(1.09–1.37)	0.0004	(1.08–1.37)	0.0002	(1.07–1.39)	0.003
Habit of exercise	0.44		0.43		0.59		0.43	
(yes)	(0.20–0.98)	0.0451	(0.17–1.10)	0.0886	(0.22–1.58)	0.2928	(0.13–1.42)	0.1645
Total energy intake per IBW	0.95		0.94		0.95		0.97	
(per 1 kcal/kg/day increase)	(0.91–1.00)	0.0262	(0.89–1.00)	0.0346	(0.89–1.03)	0.2312	(0.89–1.05)	0.3976

OR, odds ratio; CI, confidence interval; IBW, ideal body weight. Unadjusted and adjusted odds ratios (ORs) and the *p*-values were calculated using univariate and multivariate logistic regression analyses, respectively.

## Data Availability

The data presented in this study are available on request from the corresponding author.

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
