# Peer review of "The Impact of Nutritional Markers and Dietary Habits on the Bioimpedance Phase Angle in Older Individuals"

_nutrients, 2023, doi:10.3390/nu15163599_

Round 1
Reviewer 1 Report
Several key studies from the past 10 years regarding phase angle in older adults, which mainly represents the ICW/ECW ratio, are missing (here’s a recent meta analysis: Rev Endocr Metab Disord
. 2023 Jun;24(3):439-449. doi: 10.1007/s11154-022-09747-4.). I would encourage to authors to read more around the subject as these aspects need to be considered in the introduction to form a basis for this paper. Also, the rationale for the conduct of this study is not very clear. I therefore suggest revising the introduction to make clear why your study is of importance for the field.
Please provide results for a power analysis that indicate your sample size was appropriate
There is a basic need to describe the technical characteristics of the BI device. What is the calibration method to ensure validity (accuracy and precision) of the bioimpedance measurements? What is the technical error of measurement in vivo? Provide readers with a concise description of what this BI device measures. In particular, what are the measurements detected by this tool? Do they directly measure the raw bioimpedance parameters (e.g., R, Xc and phase angle)?
Most importantly, what equations were used to estimate body mass compoents? Are they equations developed using the BI device or an instrument that works with similar characteristics (frequency and technologies)?
The manuscript largely lacks illustrative representation by summarizing correlations in tables without showing the original data but rather an r/r2/p value. This is a poor representation of scientific data.
The discussion section is very descriptive and offers limited comparisons to previous research. It seems as then impact of diet on phase angle is the main practical application. Similarly, how do practitioner benefit from that? Again, the discussion section fails to relate the findings to this particular application of interest. Moreover, it is important to consider that the phase angle is a dependent instrument and that the instrumental sensitivities are different. Therefore, no comparisons can be made between studies that measure PA with different devices. Authors are therefore encouraged to make substantial changes throughout to improve the overall quality. In the current form the rationale for the study is not clear, the new value is unclear, and I have difficulties finding specific take home messages for practitioners.
Author Response
We extend our sincere gratitude to the reviewer for dedicating valuable time to review our paper. We have carefully considered and addressed all of their comments and suggestions, aiming to enhance the quality of this manuscript. We hope that the revisions made align with the reviewer's expectations and contribute to the overall improvement of the work.
COMMENT
Several key studies from the past 10 years regarding phase angle in older adults, which mainly represents the ICW/ECW ratio, are missing (here’s a recent meta analysis: Rev Endocr Metab Disord. 2023 Jun;24(3):439-449. doi: 10.1007/s11154-022-09747-4.). I would encourage to authors to read more around the subject as these aspects need to be considered in the introduction to form a basis for this paper. Also, the rationale for the conduct of this study is not very clear. I therefore suggest revising the introduction to make clear why your study is of importance for the field.
RESPONSE TO COMMENT
Thank you for your valuable feedback and providing the essential literature.
In the Introduction section, we have incorporated the suggested meta-analysis literature that highlights the positive impact of resistance exercise on PhA improvement. We have also mentioned the ICW/ECW ratio which PhA represented in the revised Introduction.
In addition, we have refined the purpose of our study to provide greater clarity.
While numerous reports have explored the relationship between PhA and nutritional status in patients with debilitating diseases, research examining the association between PhA, dietary habits, and nutritional indicators in healthy individuals remains limited. In light of this, the primary objective of this study is to investigate the correlation between PhA and dietary habits, as well as nutritional indicators, among elderly individuals undergoing health check-ups. We seek to better understand whether dietary habits contribute to the decline or improvement of PhA in healthy subjects.
COMMENT
Please provide results for a power analysis that indicate your sample size was appropriate
RESPONSE TO COMMENT
Unfortunately, prior to this cross-sectional study, we did not conduct a power analysis to estimate sample size. This is because this cross-sectional study was a pioneering, exploratory study and there was insufficient previous information on which to base a power analysis. Therefore, we cannot rule out the possibility that the significant differences detected for the principal findings in men but not in women could be attributed to the insufficient sample size.
Thank you for pointing this out. We addressed this point in the discussion section as a limitation.
COMMENT
There is a basic need to describe the technical characteristics of the BI device.
What is the calibration method to ensure validity (accuracy and precision) of the bioimpedance measurements? What is the technical error of measurement in vivo? Provide readers with a concise description of what this BI device measures. In particular, what are the measurements detected by this tool? Do they directly measure the raw bioimpedance parameters (e.g., R, Xc and phase angle)? Most importantly, what equations were used to estimate body mass compoents? Are they equations developed using the BI device or an instrument that works with similar characteristics (frequency and technologies)?
RESPONSE TO COMMENT
As per your suggestion, we have added the technical characteristics of the bioelectrical impedance (BI) device in the Methods section as below.
The body composition of the participants was evaluated using segmental multifrequency bioelectrical impedance analysis (InBody 720, InBody Japan, Tokyo, Japan). The system separately measured the impedance of the participant’s right arm, left arm, trunk, right leg, and left leg at six different frequencies (1, 5, 50, 250, 500, and 1000 kHz) for each body segment. The analyzer can directly measure fat free mass (FFM), soft lean mass (SLM), and body fat mass (BFM) without using an equation for correction. Whole-body, appendicular fat mass, and lean soft tissue mass measured by InBody 720 were found to correlate well with those measured by dual-energy X-ray absorptiometry and have been validated previously [Kim, M.; Shinkai, S.; et al. Comparison of Segmental Multifrequency Bioelectrical Impedance Analysis with Dual-Energy X-Ray Absorptiometry for the Assessment of Body Composition in a Community-Dwelling Older Population. Geriatr. Gerontol. Int. 2015, 15, 1013–1022.].
Whole-body raw values of R and Xc at 50 kHz were measured using InBody 720, and PhA values were obtained by calculating PhA (°) = [- arc tangent (Xc/R)×180°/π] .
COMMENT
The manuscript largely lacks illustrative representation by summarizing correlations in tables without showing the original data but rather an r/r2/p value. This is a poor representation of scientific data.
RESPONSE TO COMMENT
Following your valuable suggestion, we have created the necessary figures to enhance the presentation of our research findings.
Figure 1:
(a) Correlation between PhA values and levels of serum cholinesterase in males and females where r denotes the correlation coefficient.
(b) Comparison of total calorie, carbohydrate intake per IBW and protein intakes per IBW between the high- and low-PhA groups in males.
(c) Comparison of total calorie, carbohydrate intake per IBW and protein intakes per IBW between the high- and low-PhA groups in females.
COMMENT
The discussion section is very descriptive and offers limited comparisons to previous research. It seems as then impact of diet on phase angle is the main practical application. Similarly, how do practitioner benefit from that? Again, the discussion section fails to relate the findings to this particular application of interest. Moreover, it is important to consider that the phase angle is a dependent instrument and that the instrumental sensitivities are different. Therefore, no comparisons can be made between studies that measure PA with different devices. Authors are therefore encouraged to make substantial changes throughout to improve the overall quality. In the current form the rationale for the study is not clear, the new value is unclear, and I have difficulties finding specific take home messages for practitioners.
RESPONSE TO COMMENT
We sincerely appreciate your valuable feedback, which has prompted us to make significant revisions to the Discussion section.
We have refined the purpose of our study in the Introduction to ensure clarity and revised the Discussion to address the specific points of the purpose.
It is worth noting that our study's focus on healthy subjects undergoing health check-ups is a notable strength, as it allows us to examine PhA changes in the absence of debilitating diseases that may confound results in other studies. This approach provides a unique suggest that preventing a shortage of carbohydrates and ensuring sufficient intake of total calories may contribute to increased PhA, promoting healthy aging among healthy older males.
In addition, based on the essential received feedback that it is important to consider that the phase angle is a dependent instrument and that the instrumental sensitivities are different, we have cited literature from studies that used the same InBody 720 device to report the absolute values (mean) of PhA in the Introduction section. Additionally, we established the rationale for the cutoff value used to divide participants into two groups based on a study that utilized the Tanita device. This study also involved Japanese individuals aged 65 years and above the same as our study, and the reported average PhA for males was 5.33±0.62 and for females was 4.66±0.5, which closely resembled the mean values obtained in our study. As a result, we adopted the same cutoff value for our research.
Reviewer 2 Report
The manuscript titled “The Impact of Nutritional Markers and Dietary Habits on the 2 Bioimpedance Phase Angle in Older Individuals”.
Introduction
The introduction should be improved.
Methods
A robust list of biochemical markers.
I have a few questions about this:
Were the biochemical analyses performed in fasting or not?
What is the time of blood collection and biochemical analysis performed?
For the number of biochemical markers analysed, the interaction between the markers.
The discussion should also improve.
The authors have a robust list of analysed markers, this is an asset to the manuscript.
Discussion/Conclusion
The Discussion and Conclusion should be improved.
References
Few references and low impact, please update this.
References are out of date.
Please update this, at least reference the last 5-10 years.
Author Response
We extend our sincere gratitude to the reviewer for dedicating valuable time to review our paper. We have carefully considered and addressed all of their comments and suggestions, aiming to enhance the quality of this manuscript. We hope that the revisions made align with the reviewer's expectations and contribute to the overall improvement of the work.
COMMENT
Introduction
The introduction should be improved.
RESPONSE TO COMMENT
We have refined the purpose of our study in the Introduction to provide greater clarity.
While numerous reports have explored the relationship between PhA and nutritional status in patients with debilitating diseases, research examining the association between PhA, dietary habits, and nutritional indicators in healthy individuals remains limited. In light of this, the primary objective of this study is to investigate the correlation between PhA and dietary habits, as well as nutritional indicators, among elderly individuals undergoing health check-ups. We seek to better understand whether dietary habits contribute to the decline or improvement of PhA in healthy subjects.
COMMENT
Methods
A robust list of biochemical markers.
I have a few questions about this:
Were the biochemical analyses performed in fasting or not?
What is the time of blood collection and biochemical analysis performed?
RESPONSE TO COMMENT
The examinations were conducted in the morning after an overnight fast.
We have added this information to the Method section of our paper.
COMMENT
For the number of biochemical markers analysed, the interaction between the markers.
RESPONSE TO COMMENT
Thank you for pointing this out. Here we present the results of the Pearson correlation coefficient analysis, which was used to assess the correlations between markers of nutritional indices.Correlation between serum Alb and ChE: r=0.124 (p=0.1536)
Correlation between total lymphocyte count and serum ChE: r=-0.0075 (p=0.9317)
Correlation between total lymphocyte count and serum Alb: r= 0.0171 (p=0.8447)
COMMENT
The discussion should also improve.
The authors have a robust list of analysed markers, this is an asset to the manuscript.
Discussion/Conclusion
The Discussion and Conclusion should be improved.
RESPONSE TO COMMENT
We have refined the purpose of our study in the Introduction to ensure clarity and revised the Discussion to address the specific points of the purpose.
It is worth noting that our study's focus on healthy subjects undergoing health check-ups is a notable strength, as it allows us to examine PhA changes in the absence of debilitating diseases that may confound results in other studies. This approach provides a unique suggest that preventing a shortage of carbohydrates and ensuring sufficient intake of total calories may contribute to increased PhA, promoting healthy aging among healthy older males.
In addition, thank you for your valuable feedback about the strength of our study. We have taken your suggestion into consideration and added a discussion in the manuscript to highlight the robust list of analysed markers as a significant asset to our research.
COMMENT
References
Few references and low impact, please update this.
References are out of date.
Please update this, at least reference the last 5-10 years.
RESPONSE TO COMMENT
Thank you for pointing this out. We have reviewed all the references and updated the citations to include as many references as possible that are within the last 10 years. We highlighted the revised references in the manuscript.
Round 2
Reviewer 1 Report
The authors did not consider the suggested references for improving the introduction sections. However, more info has been added and this makes the rationale of the study more clear.
It would be interesting to discuss the other qualitative BIA-based approach for assessing body composition through bioimpedance analysis. In this regard, a recent study presented new references and highlighted the potential of the BIVA method: New bioelectrical impedance vector references and phase angle centile curves in 4,367 adults: the need for an urgent update after 30 years. DOI:https://doi.org/10.1016/j.clnu.2023.07.025. Therefore, my suggestion is to promote the phase angle and the BIVA approaches making clear to the readers the possibility of using alternative ways than the conventional BIA-based quantitative method.
Author Response
We deeply value your insightful feedback. Based on your valuable suggestion, we have incorporated a discussion of an additional qualitative approach for assessing body composition utilizing BIA. We also wanted to acknowledge the valuable contribution of the referenced study (DOI: https://doi.org/10.1016/j.clnu.2023.07.025) in our discussion.
Page9, Line 261
Furthermore, it is crucial to acknowledge that PhA values exhibit variability attributed to race, gender, age, and body size. As a result, meticulous attention is essential when interpreting these outcomes. As an alternative approach, Bioelectrical Impedance Vector Analysis (BIVA) employs the normalization of impedance parameters R and Xc with respect to height, utilizing a bivariate vector on the RXc graph [39]. BIVA provides a qualitative assessment of soft tissue that remains unaffected by body size, offering valuable insights into hydration status and the integrity of cellular mass [6]. Another innovative strategy involves standardizing PhA using reference values adjusted for age, sex, and BMI. Standardized values potentially provide a more reliable indication of nutritional or health status in comparison to absolute PhA values [6,39].
6. Bioelectrical phase angle and impedance vector analysis--clinical relevance and applicability of impedance parameters.
Norman K, Stobäus N, Pirlich M, Bosy-Westphal A.
Clin Nutr. 2012 Dec;31(6):854-61.
39. New bioelectrical impedance vector references and phase angle centile curves in 4,367 adults: The need for an urgent update after 30 years.Campa F, Coratella G, Cerullo G, Stagi S, Paoli S, Marini S, Grigoletto A, Moroni A, Petri C, Andreoli A, Ceolin C, Degan R, Izzicupo P, Sergi G, Mascherini G, Micheletti Cremasco M, Marini E, Toselli S, Moro T, Paoli A. Clin Nutr. 2023 Jul 31;42(9):1749-1758.
Reviewer 2 Report
No further comments.
Author Response
Thank you very much for reviewing our manuscript and for your time and effort in providing feedback.